Piwi1 is essential for gametogenesis in mollusk Chlamys farreri

Ma Xiaoshi
Ji Aichang
Zhang Zhifeng
Yang Dandan
Liang Shaoshuai
Wang Yuhan
Qin Zhenkui qinzk@ouc.edu.cn
Key Laboratory of Marine Genetics and Breeding (Ocean University of China), Ministry of Education, Ocean University of China , Qingdao , China
Brogna Saverio
Electronic publication date: 2017 Jun 23
Publication date: 2017
Volume: 5
Electronic Location ID: e3412
Received 2017 Mar 28; Accepted 2017 May 14
Copyright: ©2017 Ma et al.
Copyright year: 2017
Copyright holder: Ma et al.
License: This is an open access article distributed under the terms of the Creative Commons Attribution License, which permits unrestricted use, distribution, reproduction and adaptation in any medium and for any purpose provided that it is properly attributed. For attribution, the original author(s), title, publication source (PeerJ) and either DOI or URL of the article must be cited.
License URL: https://creativecommons.org/licenses/by/4.0/

Keywords: Chlamys farreri, Cf-piwi1, Germ cell, Gametogenesis, Testis, Ovary

Funding: Laboratory for Marine Fisheries and Aquaculture, Qingdao National Laboratory for Marine Science and Technology 2015ASKJ02 National Natural Science Foundation of China 31602139 This work was supported by the Scientific and Technological Innovation Project from Laboratory for Marine Fisheries and Aquaculture, Qingdao National Laboratory for Marine Science and Technology (2015ASKJ02) and National Natural Science Foundation of China (31602139). The funders had no role in study design, data collection and analysis, decision to publish, or preparation of the manuscript.

==============================
Piwi (P-element induced wimpy testis) is an important gene involved in stem cell maintenance and gametogenesis in vertebrates. However, in most invertebrates, especially mollusks, the function of Piwi during gametogenesis remains largely unclear. To further understand the function of Piwi during gametogenesis, full-length cDNA of Piwi1 from scallop Chlamys farreri (Cf-Piwi1) was characterized, which consisted of a 2,637 bp open reading frame encoding an 878-amino acid protein. Cf-Piwi1 mRNA was mainly localized in the spermatogonia, spermatocytes, oogonia, oocytes of early development and intra-gonadal somatic cells. Additionally, the knockdown of Cf-Piwi1 by injection of Cf-Piwi1-dsRNA (double-stranded RNA) into scallop adductor led to a loss of germ cells in C. farreri gonads. Apoptosis was observed mainly in spermatocytes and oocytes of early development, as well as in a small number of spermatogonia and oogonia. Our findings indicate that Cf-Piwi1 is essential for gametogenesis in the scallop C. farreri.

Introduction

Gametogenesis is the basis of animal reproduction and mainly includes germ stem cell self-renewal, meiosis and maturation of gametes. Studies of gametogenesis will help improve the reproductive ability and preservation of economically important species. It is known that many genes are involved in this process for model organisms, but the roles of these genes remain largely unknown in non-model organisms.

Piwi (P-element induced wimpy testis), a PIWI subfamily member of the Argonaute superfamily, is identified based on two conserved domains, PAZ and PIWI (Cerutti, Mian & Bateman, 2000). The PAZ domain, at the center of the amino acid sequence, contains a typical single stranded nucleic acid binding motif that can bind to the 3′ end of short RNA (Lingel et al., 2003; Yan et al., 2003). The PIWI domain, found in the C-terminal region, functions to maintain Piwi’s stability and is structurally similar to the RNase H catalytic domain (Liu et al., 2004; Song et al., 2004).

The Piwi gene was first identified in Drosophila melanogaster and demonstrated a potentially important role in maintaining germ cells (GCs) (Lin & Spradling, 1997). Subsequently, Piwi homologues were reported in a variety of species, including Caenorhabditis elegans, Bombyx mori, Danio rerio, Coturnix coturnix and Homo sapiens (Lau et al., 2001; Sasaki et al., 2003; Houwing, Berezikov & Ketting, 2008; Chen et al., 2012; Tatsuke et al., 2014). The expression of the Piwi gene is mostly restricted to gametogenesis and early embryonic development, but its expression pattern and functions are not consistent in different animals (Deng & Lin, 2002; Megosh et al., 2006; Carmell et al., 2007; Houwing, Berezikov & Ketting, 2008; Wang & Reinke, 2008). In D. melanogaster, Piwi mutants eliminate the self-renewing division of germ stem cells (GSCs), and overexpressing Piwi in the germarium somatic cells results in an increase in number of GSCs and the rate of mitosis (Cox et al., 1998). In the flatworm Macrostomum lignano, knockdown of Piwi results in a complete elimination of all stem cells, including GSCs and somatic stem cells (De et al., 2009). Tatsuke et al. (2014) suggested that Siwi (the silkworm homologue of the Piwi protein) recruits HP1 proteins to a target site guided by the Piwi-piRNA complex, and then the Piwi-HP1 complex functions as a rapid transcriptional repressor to regulate gene expression in B. mori.

Mollusks are one of the most abundant and biologically diverse groups in the animal kingdom. Identifying early GCs is beneficial to the study of gametogenesis. However, it is difficult in histological sections of the mollusk gonads to distinguish various types of germ cells accurately, especially for the early development stages, because some features, such as cell size and karyoplasmic ratio, are diverse in different sections. These problems limit the study on the molecular mechanism of gametogenesis in mollusks. In this study, we cloned full-length cDNA of Piwi1 in the scallop Chlamys farreri, a commercially important bivalve mollusk in China, and revealed its expression characteristics in the gonads during gametogenesis. Furthermore, the role of C. farreri Piwi1 (Cf-Piwi1) in the scallop during gametogenesis was examined using RNAi. Our aims are to demonstrate the function of Cf-Piwi1 during gametogenesis and investigate its potential feasibility as a molecular marker to identify early GCs in the scallop gonads.

Materials and Methods

Ethics statement

The collection and handing of the scallops C. farreri were performed in accordance with the Institutional Animal Care and Use Committee of the Ocean University of China and the local government.

Specimen collection and sampling

Adult C. farreri scallops with a mean shell height of 6.28 ± 0.43 cm were collected from Shazikou (Qingdao, China). Gonads were dissected into 0.2 cm3 pieces. Some of these pieces were fixed in 4% paraformaldehyde in 0.1 M phosphate buffer (pH 7.4) at 4 °C for 24 h, dehydrated through serial methanol dilutions (25, 50, 75 and 100%) and stored in pure methanol at −20 °C for in situ hybridization (ISH). Some other pieces were fixed in Bouin’s solution (picric acid, saturated aqueous solution - 75 ml; formalin, 40% aqueous solution - 25 ml; acetic acid, glacial - 5 ml) for 24 h and then stored in 70% ethanol for histological observation. The remaining pieces were immediately frozen in liquid nitrogen and stored at −80 °C for total RNA isolation. All the reagents used without specific indication were provided by Sangon Biotech (Shanghai, China).

Histology

Gonads stored in 70% ethanol were dehydrated in an ethanol dilution series, cleared with xylene, and embedded in paraffin wax according to the description of Liu et al. (2014). Sections were made at 5 µm thickness and stained with hematoxylin and eosin. Observations and digital images were taken with a Nikon E80i microscope (Nikon, Tokyo, Japan).

Gonads were divided into four stages according to previously described morphological characteristics (Liu et al., 2012). The gonadosomatic indices (GSI = gonad weight/soft tissue body weight × 100%) are defined as resting stage (GSI 3.73% for females and 3.49% for males), proliferative stage (GSI 4.32% for females and 4.38% for males), growing stage (GSI 5.39% for females and 5.42% for males) and mature stage (GSI 14.29% for females and 12.48% for males).

Total RNA extraction and reverse transcription

Total RNA was extracted using the thiocyanate–phenol–chloroform method according to Chomczynski & Sacchi (1987). Quality and quantity of the RNA were measured using agarose gel electrophoresis and spectrophotometry. Reverse transcription for full-length cDNA cloning and qRT-PCR were performed according to manufacturer instructions using the SMARTer™ RACE cDNA Amplification Kit (Clontech, Mountain View, USA) and Primescript™ RT reagent Kit with gDNA Eraser (Perfect Real Time) (Takara, Otsu, Japan), respectively.

Cloning and sequence analysis of full-length cDNA

A Piwi1 cDNA fragment of 311 bp was obtained from the C. farreri transcriptome (Wang et al., 2013) and compared to the National Center for Biotechnology Information (NCBI) database using BLASTX. Amplification of 5′- and 3′-RACE were conducted with scallop testis cDNA and two specific PCR primers (PR-5′: 5′-GCAACAGACATCAA CATCTGTTTCTTGG-3′, PR-3′: 5′-ATGCTGATTGGAGCAGAGATCTTCGTGG-3′) according to the SMART™ RACE cDNA Amplification Kit protocol (Clontech, Mountain View, USA). PCR products were gel-purified and cloned into the pMD18-T vector (Takara, Otsu, Japan) then transformed into Escherichia coli DH5α competent cells (Takara, Otsu, Japan). Positive clones were selected and sequenced. The full-length cDNA sequence was assembled using DNASTAR, Lasergene version 7.1.

The identity and similarity of the deduced amino acid sequence were analyzed with other known PIWI1 (Homo sapiens, Mus musculus, Sus scrofa, Gallus gallus, Caprimulgus carolinensis, Xenopus tropicalis, Danio rerio, Alitta virens, Lottia gigantea, Crossostrea gigas, Mytilus galloprovincialis, Caenorhabditis elegans) in GenBank using the online BLASTX tool. Multiple alignments were performed using the software CLUSTALX version 1.81 and DNAMAN version 8.0. We conducted a phylogenetic analysis using the neighbor-joining method in MEGA 5.0 with 1,000 bootstrap replicates (Koichiro et al., 2011).

qRT-PCR analysis

qRT-PCR was conducted using SYBR Green Real-Time PCR Master Mix (TOYOBO, Osaka, Japan) and an ABI 7500 Real-Time PCR System (Applied Biosystems, Foster City, USA). A parallel amplification of the C. farreri elongation factor 1α (EF-1α) reference transcript (GenBank accession no. AEX08674.1) was performed to normalize expression data of Cf-Piwi1 transcripts. Two pairs of specific primers, Piwi1 F-1: 5′-CGGAGGCGTT GTGTGTAGCA-3′, Piwi1 R-1: 5′-CTGTCCATCCCCAACACCATC-3′ for amplifying a 193 bp of Cf-Piwi1, and EF-1α F: 5′-ATCCTTCCTCCATCTCGTCCT-3′, EF-1α R: 5′-GGCACAGTTCCAATACCTCCA-3′ for amplifying an 86 bp of Cf-EF-1α were designed. RT-PCR conditions consisted of an initial denaturation step at 95 °C for 30 s followed by 40 cycles of 5 s of denaturation at 95 °C and 30 s of annealing and extension at 60 °C. Gonads from five individuals at each developmental stage were sampled, and triplicate assays for each gonad cDNA were conducted. The data were analyzed using the ABI 7500 system SDS software version 1.4 with automatically set baseline and cycle threshold values. Relative Cf-Piwi1 mRNA levels were calculated based on the 2−ΔΔCt method.

All data are presented as the means ± standard error of five samples with three parallel repetitions. Differences between the means were tested using one-way analysis of variance (ANOVA) followed by least significant difference tests with the significance level set at P < 0.05 in SPSS version 17.0.

Tissue ISH

DIG-labeled RNA sense and anti-sense probes were synthesized from a 557 bp fragment of Cf-Piwi1 from sites 3,078 to 3,634 according to instructions with the DIG RNA Labeling kit (Roche, Basel, Switzerland). Gonads stored for tissue ISH were cleared in xylene and embedded in paraffin wax before sectioning at 5 µm for testes and 7 µm for ovaries. Sections were fixed to a slide with 0.1% polylysine for 10 h at 37 °C. Before washing three times with PBST (phosphate-buffered saline with 0.1% Tween 20) and digesting with proteinase K (2 µg ml−1) for 15 min at 37 °C, samples were dewaxed in xylene and rehydrated through a descending series of methanol dilutions. After fixing with 4% paraformaldehyde for 1 h and prehybridizing at 60 °C for 6 h in hybridization buffer (50% formamide, 5% SSC, 5 mM EDTA, 100 mg ml−1 ribonucleic acid, 1.5% blocking reagent, 0.1% Tween 20), samples were hybridized with digoxigenin (DIG)-labeled probes at 1 mg ml−1 in hybridization buffer for 16 h at 60 °C. Following hybridization, samples were washed in maleic acid buffer (0.1 M maleic acid, 0.15 M NaCl, 0.1% Tween 20, pH 7.5) and incubated with alkaline phosphatase-conjugated anti-DIG antibody from the DIG Nucleic Acid Detection Kit (Roche, Basel, Switzerland) at 4 °C. After staining with NBT/BCIP (Roche, Basel, Switzerland) for 3 h at room temperature, the sections were counterstained with 1% neutral red. Hybridization signals were detected and photographed using a Nikon E80i microscope (Nikon, Tokyo, Japan).

dsRNA synthesis

A 726 bp fragment of Cf-Piwi1 cDNA from sites 11 to 736 was amplified using the primers Piwi1 F-2: 5′-TAATACGACTCACTATAGGGTTGAGAGGCAAGAAGTAACA-3′ and Piwi1 R-2: 5′-TAATACGACTCACTATAGGGGTACAGATGAAGGCACTGTG-3′ (T7 promoter sequence underlined) with C. farreri testis RNA as the template. The purified PCR fragment was transcribed, and the double-strand RNA (dsRNA) was synthesized in vitro using T7 MEGAscript RNAi Kits (Ambion, Austin, USA) according to manufacturer instructions. Quality and quantity of the Piwi1-dsRNA was measured by 1% agarose gel electrophoresis and spectrophotometry.

dsRNA injection and sampling

Scallops with a mean shell height of 6.13 ± 0.54 cm at the proliferative stage were collected from Shazikou (Qingdao, China) and maintained in aerated running filtered seawater and fed with single cell algae during the experiment. A total of 75 scallops were randomly assigned to three groups. Scallops from the dsRNA group and PBS group were injected with 25 µg Cf-Piwi1 dsRNA diluted in 100 µl PBS and 100 µl PBS only into adductor muscle, respectively. Scallops in the blank group were not injected with anything. Two injections were conducted during the experiment: at the beginning and at day 7 of the experiment. On the third day after injection, three scallops from each group were removed randomly, and their gonads were sampled as described above to estimate the Cf-Piwi1 knockdown effect. On the 10th day, eight scallops (five males and three females) from each group were sampled again.

Types of germ cell quantification

Five-µm gonadal sections were conducted from scallops of each group following the method mentioned in Histology. To determine the effect of Cf-Piwi1 knockdown, five squares (6,400 µm2 for ovary and 1,600 µm2 for testis) of the sections were randomly picked to calculate the mean number and composition of germ cells in the gonads of each group. Differences between cell quantities of different groups were tested using one-way analysis of variance (ANOVA) followed by least significant difference tests with the significance level set at P < 0.05 in SPSS version 17.0.

TUNEL assay

Five-µm sections of the gonads were prepared after the RNAi experiment. To detect in situ cell apoptosis, a terminal deoxynucleotidyl transferase-mediated dUTP nick end labeling (TUNEL) assay was performed using a TdT-mediated dUTP apoptosis detection kit (Promega, Madison, USA) with a hematoxylin counterstain. The sections were observed and photographed using a Nikon E80i microscope (Nikon, Tokyo, Japan).

Results

Sequence and characteristics of the Cf-Piwi1 full-length cDNA

The full length of the Cf-Piwi1 cDNA was 4,986 bp (GenBank accession number: KR869093) with a 59 bp 5′ untranslated region (UTR), a 2,290 bp 3′ UTR and a 2,637 bp open reading frame (ORF), encoding a putative protein of 879 amino acids, with a predicted molecular weight of 99.34 kDa and theoretical isoelectric point of 9.16. Multiple alignment indicated that the predicted protein contained a PAZ domain and a PIWI domain (Fig. S1) and was highly homologous to other known PIWI1, with 57% identity to Crassostrea gigas, 51% to Danio rerio and 53% to Homo sapiens. Phylogenetic analysis showed that the predicted Cf-Piwi1 first clustered with C. gigas and M. galloprovincialis, and then followed established evolutionary hypotheses (Fig. 1).

Figure 1 Phylogenetic analysis of Piwi1 among various species based on the multiple sequence alignment.

Quantitative expression of Cf-Piwi1 mRNA in C. farreri gonads during the reproductive cycle

Expression levels of Cf-Piwi1 in C. farreri gonads increased significantly from the resting to the mature stage (Fig. 2A and 2B). Cf-Piwi1 expression in testes at the mature stage was approximately 2.5 times higher than that of the resting stage. In ovaries, it was about two-fold higher at the mature stage than the resting stage. No significant differences in Cf-Piwi1 expression levels were observed between the ovary and testis at the same developmental stages (P > 0.05).

Figure 2 Relative abundance and location of Cf-Piwi1 mRNA in C. farreri gonads.

Relative abundance of Cf-Piwi1 mRNA detected by qRT-PCR in (A) ovary and (B) testis. The expression level in gonads at the resting stage was set as 1.00; Values are the mean ± SEM; n = 3; Different letters indicate statistically significant differences (P < 0.05). (C). Location of Cf-Piwi1 mRNA detected by tissue ISH. Negative ISH using a sense probe (b, e, i). Positive signal from the anti-sense probe is stained in dark blue. (a), (c), (d) and (f), Ovaries at the resting, proliferative, growing, and mature stage, respectively; (g), (h), (j) and (k), Testes at the resting, proliferative, growing, and mature, respectively; (l), a different sight of the same section of (k) under microscope. ISC, Intragonadal somatic cell; Moc, Mature oocyte; Og, Oogonium; Oc, Oocyte; Sg, Spermatogonium; Sc, Spermatocyte; St, Spermatid; Sz, Spermatozoon. Magnification: Bar is 20 µm.

Cytolocation of Cf-Piwi1 mRNA in C. farreri gonads during gametogenesis

Cf-Piwi1 mRNA was mainly located in GCs during early development. In ovaries, Cf-Piwi1 transcripts were detected in oogonia, oocytes of early development and intra-gonadal somatic cells (ISCs) of germinal acini, but no positive signal was detected in mature oocytes (Figs. 2C–2H). In testes, obvious positive signals were observed in spermatogonia, spermatocytes and ISCs of germinal acini. However, no positive signal was visible in spermatids and spermatozoa (Figs. 2C, 2I –2N). Moreover, no positive signal was detected in gonads using sense probes (Figs. 2D, 2G and 2K).

Cf-Piwi1 knockdown led to abnormal development and apoptosis of GCs

qRT-PCR detected that levels of the Cf-Piwi1 mRNA decreased significantly in the gonads of the dsRNA group than that of the PBS and Blank groups. The reduction of Cf-Piwi1 expression levels in the C. farreri gonads between the first injection and the second injection in the dsRNA group was very similar, and the declines in the ovaries and testes were approximately 30% or 35% of that in the blank group, respectively (Fig. 3A and 3B).

Figure 3 Expression of Cf-Piwi1 mRNA and histology of scallop gonads after RNAi.

Relative experiment level of Cf-Piwi1 mRNA detected by qRT-PCR in (A) ovary and (B) testis on 10th day after RNAi. The expression level in gonads of the blank group was set as 1.00; Values are the mean ± SEM; n = 5 in the testes; n = 3 in the ovaries; Different letters indicate statistically significant differences (P < 0.05). (C). Histological observation of scallop gonads on 10th day after RNAi. (a) and (d), ovary in the blank group; (b) and (e), ovary in the PBS group; (c) and (f), ovary in the dsRNA group; (g) and (j), testis in the blank group; (h) and (k), testis in the PBS group; (i) and (l), testis in the dsRNA group. Dc, Darkly stained cell; Og, Oogonium; Oc, Oocyte; Sc, Spermatocyte; Sg, Spermatogonium; St, Spermatid; Sz, Spermatozoon. Magnification: Bar is 40 µm for (a), (b), (c), (g), (h) and (i); Bar is 20 µm for (d), (e), (f), (j), (k) and (l).

To investigate the effects of Cf-Piwi1 deficiency on gametogenesis, we performed a histological analysis. Cf-Piwi1-dsRNA provoked several defects in the development of GCs in both testes and ovaries. Compared with scallops from control groups, most oocytes in the ovaries of dsRNA scallops were at early developmental stage and many of them were stained darkly and presented abnormal morphological characteristics, implying that Cf-Piwi1 downregulation might inhibit oocyte development (Figs. 3C–3H). In the testes of the Cf-Piwi1 knockdown scallops, the arrangement of GCs in the germinal acini became loose, spermatids occurred only in few germinal acini and the number of spermatids was smaller compared visually to that of the blank and PBS groups (Figs. 3I–3N). Furthermore, we quantified the number and composition of germ cells in each group. In ovaries, more than half the number of all kinds of germ cells decreased after knocking down of Cf-Piwi1, but the proportion of Cf-Piwi1 expression cells (ISC 36.3%, oogonia 30.5% and oocyte 33.2%) increased when compared with control groups (ISC 32.1%, oogonia 35.7% and oocyte 18.2%) (Fig. 4A, Table S1). Similar results were obtained in testis with the percentage of spermatogonia and spermatocyte changed from 11.4% and 67.2% of the control groups to that of 38.2% and 54.5% after RNAi (Fig. 4B, Table S2). Interestingly, we also found that in some germinal acini of the Cf-Piwi1 knockdown scallops the number of spermatocytes greatly decreased while spermatogonia and spermatids persisted (Figs. 3K and 3N).

Figure 4 Quantification of the germ cells in scallop gonads after Cf-Piwi1 knockdown and cell apoptosis analysis.

Quantification of germ cells in (A) ovary, (B) testis and (C) cell apoptosis detection on 10th day after RNAi. (a) and (d), ovary in the blank group; (b) and (e), ovary in the PBS group; (c) and (f), ovary in the dsRNA group; (g) and (j), testis in the blank group; (h) and (k), testis in the PBS group; (i) and (l), testis in the dsRNA group. Apo, apoptosis cell; ISC, Intragonadal somatic cell; Moc, Mature oocyte; Og, Oogonium; Oc, Oocyte; Sg, Spermatogonium; Sc, Spermatocyte; St, Spermatid; Sz, Spermatozoon. The values are the mean ± SEM; n = 5 in both testes and ovaries; different letters indicate statistically significant differences (P < 0.05). Magnification: Bar is 40 µm for (a), (b), (c), (g), (h) and (i); Bar is 20 µm for (d), (e), (f), (j), (k) and (l).

The TUNEL assay results revealed that some of the oogonia and many oocytes were in the process of apoptosis in the ovaries of the dsRNA scallops, and that the majority of spermatocytes and partial spermatogonia in Cf-Piwi1 knockdown testes were undergoing apoptosis (Figs. 4E, 4H, 4K and 4N). Few apoptotic cells were found in the gonads of the blank and PBS groups (Figs. 4C, 4D, 4F, 4G, 4I, 4J, 4L and 4M).

Discussion

Cf-Piwi1 expression pattern in GCs is similar to that of fish

Localizations of Piwi transcripts are diverse in gonads of different species, although they are known to express mainly in GCs. In the planarian Schmidtea mediterranea, Piwi mRNA is visible in somatic stem cells and GCs (Reddien et al., 2005; Rossi et al., 2006; Palakodeti et al., 2008). In D. melanogaster, Piwi is expressed in all the cells of gonads (Cox, Chao & Lin, 2000). In D. renio, Ziwi, a Piwi homologue, is found only in GCs of gonads, where its expression appears to be the strongest in GCs at the mitotic and early meiotic stages (Houwing, Berezikov & Ketting, 2008). Similarly, in medaka (Oryzias latipes), Piwi is expressed in spermatogonia, spermatocytes and all ovarian GCs (Li et al., 2012). In M. musculus, Miwi expression appears to be restricted to the primary spermatocytes, secondary spermatocytes and the elongating spermatids, and no expression is observed in somatic cells of testis (Deng & Lin, 2002). In this study, Cf-Piwi1 mRNA was expressed in male and female GCs of early development, which differs from that in mammals but is similar to that in fish.

Cf-Piwi1 is possibly a molecular marker for early GCs

Yano, Suzuki & Yoshizaki (2008) reported that rtili, a homolog of Piwi in the rainbow trout Oncorhynchus mykiss, is expressed specifically in spermatogonia and is used as a molecular marker to identify spermatogonia. In this study, we found that Cf-Piwi1 expression was not only restricted to spermatogonia, but was also specifically visible in the GCs of early development, such as spermatogonia, spermatocytes, oogonia, and oocytes of early development. Thus, it can be potentially used to identify the GCs of early development in the testes and ovaries of C. farreri.

Cf-Piwi1 is essential for gametogenesis in C. farreri

In model animals, the roles of Piwi on gametogenesis are diverse, but Piwi defects always result in the loss of GCs, reductions in nurse cell, poorly-developed egg chambers, and complete female sterility (Lin & Spradling, 1997). Cox et al. (1998) reported that Piwi mutations in D. melanogaster cause loss of GCs, but no dead cells were detected, which implies that the loss of Piwi can eliminate the self-renewing division of GSCs. Similarly, in C. elegans, decreasing Piwi expression by RNAi reduces the proliferation of GSC-equivalent cells (Cox et al., 1998). Moreover, in Zili (Ziwi-like) mutant zebrafish, almost all GCs are lost yet no apoptosis is present, suggesting that loss is possibly due to their inability to proliferate and differentiate (Houwing, Berezikov & Ketting, 2008). Mutation of a hypomorphic Zili allele blocks oogenesis in Meiosis I and induces terminal female sterility (Houwing, Berezikov & Ketting, 2008). However, Houwing et al. (2007) found that the reduction of Ziwi in D. renio leads to various spermatogenic cell losses by apoptosis. In M. musculus, significant numbers of apoptotic cells were detected in spermatocyte layers due to the loss of Mili (Kuramochi-Miyagawa et al., 2004). Miwi-knockout mice display a drastic increase in apoptotic cell numbers of testes and spermatogenic arrest at the round spermatid stage (Deng & Lin, 2002). Additionally, Miwi2 mutants exhibit spermatogenic cell apoptosis and predominantly arrest at the leptotene stage of meiosis (Carmell et al., 2007).

In this study, we found that knockdown of Cf-piwi1 lead to significant cell number reduction and most of the remaining germ cells detained at the early development stages, implying its important role in the germ cell proliferation and differentiation. In the meantime, the proportion of Cf-piwi1 expression cells increased after RNA interference also indicated that gametogenic arrest occurred at early development stages, which was accordant with the reported studies. Cell apoptosis assay presented that marked apoptosis occurred mainly in the spermatocytes of the testes and oocytes of the ovaries in Cf-Piwi1 knockdown scallops, respectively, indicating that germ cells at earlier development stages gradually died after earlier accumulation. In addition, small numbers of spermatogonia and spermatids existed despite a great reduction of spermatocytes also demonstrating that spermatogenesis was blocked in spermatocytes. All these results suggest that Cf-Piwi1 plays an important role in C. farreri gametogenesis.

Supplemental Information

Figure S1 Multiple alignment of Piwi1

Two boxes show the PAZ (N-terminal side) and Piwi (C-terminal side) domains. The identical and similar residues are highlighted in black and gray, respectively.

Click here for additional data file.

Table S1 Quantification of germ cells in five different histological sections of C. farreri ovary after RNAi

Click here for additional data file.

Table S2 Quantification of germ cells in five different histological sections of C. farreri testis after RNAi

Click here for additional data file.

Additional Information and Declarations

Competing Interests

Author Contributions

DNA Deposition

Data Availability

The authors declare there are no competing interests.

Xiaoshi Ma performed the experiments, analyzed the data, wrote the paper, prepared figures and/or tables.

Aichang Ji performed the experiments, analyzed the data.

Zhifeng Zhang conceived and designed the experiments, reviewed drafts of the paper.

Dandan Yang performed the experiments, contributed reagents/materials/analysis tools.

Shaoshuai Liang and Yuhan Wang contributed reagents/materials/analysis tools.

Zhenkui Qin reviewed drafts of the paper.

The following information was supplied regarding the deposition of DNA sequences:

The Cf-Piwi1 cDNA seqence described here are accessible via GenBank accession number KR869093.

The following information was supplied regarding data availability:

The raw data has been supplied as Supplemental Files.

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
