# Peer review of "Piwi1 is essential for gametogenesis in mollusk Chlamys farreri"

_PeerJ, doi:10.7717/peerj.3412_

## Round 0.1 · original submission · Minor Revisions

· Academic Editor

Minor Revisions

Sorry for the delay but it was difficult to find suitable reviewers. But as you can see all 3 reviewers like the manuscript. Please respond to their comments.

Reviewer 1 ·

Basic reporting

no comment

Experimental design

no comment

Validity of the findings

no comment

Additional comments

The manuscript of Ma et al., entitled "Piwi1 is essential for gametogenesis in mollusk 1 Chlamys farreri" (#17033), reports the function of Piwi1 in mollusks. Piwi1 is an important house-keeping gene in vertebrates, but in most invertebrates including mollusks, its function during gametogenesis remains unknown. The authors found out that knocking down Cf-Piwi1 gene in scallops would cause remarkable apoptosis, which occurred mainly in the spermatocytes of the testes, indicating that Cf-Piwi1 is essential for gametogenesis in the scallop C. farreri. The results are significant and well presented, and this work should provide valuable information for future researches, especially on non-model organisms.

Some minor revisions need to be done before publish.

Line 60, 'benefit' is supposed to be 'beneficial'
Line 80, 'Some other parts' rather than 'Another parts' is preferred
Line 130, abbreviation 'SEM' was not defined, could use 'standard error' here
Line 251, 'but was' add 'also'

Reviewer 2 ·

Basic reporting

The manuscript by Ma et al describes the outcome of experiments designed to show that a) the bivalve mollusc Chlamys farreri expresses a gene orthologous to the mammalian Piwi gene (a member of the argonaute family) and b) that this gene is essential for gametogenesis. The work is well written, with just a few grammatical errors that can easily be rectified. The background introductory information is adequate and the work is structured appropriately for an original scientific report.

Experimental design

The experimental design is a mix of molecular histological experiments aimed at demonstrating the expression of Piwi1 RNA in the mollusk (like Mytilus) using an in situ hybridization based approach with RNA probes and also a real-time PCR approach providing relative (to a housekeeping gene) quantitation of Piwi1 RNA in control animals and animals injected with a dsRNA derived from the Piwi1 sequence where a knock down of Piwi transcription is observed. Finally, the authors show that apoptosis of germ cells, particularly in the female is probably responsible for the loss of mature gametes in knock down animals. The research complies fully with that expected of an original report and is structured accordingly.

Validity of the findings

I believe the authors findings are valid enough although I would have preferred the injection of a sham dsRNA control in the knock down experiments rather than a PBS control as the reduction in expression of Piwi1observed was certainly not great and could conceivably have been an off target effect. The authors also need to indicate what statistical tests they applied to their data.

Additional comments

The manuscript by Ma et al describes the outcome of experiments designed to show that a) the bivalve mollusc Chlamys farreri expresses a gene orthologous to the mammalian Piwi gene (a member of the argonaute family) and b) that this gene (like its mammalian orthologue) is essential for gametogenesis. The work is well written, with just a few grammatical errors that can easily be rectified. The background introductory information is adequate and the work is structured appropriately for an original scientific report. The experimental design is a mix of molecular histological experiments aimed at demonstrating the expression of Piwi RNA in the mollusk (like Mytilus) using an in situ hybridization based approach with RNA probes and also a real-time PCR approach providing relative (to a housekeeping gene) quantitation of Piwi1 RNA in control animals and animals injected with a dsRNA derived from the Piwi sequence where a knock down of Piwi1 transcription is observed. Finally, the authors show that apoptosis of germ cells, particularly in the female is probably responsible for the loss of mature gametes in knock down animals. The research complies fully with that expected of an original report and is structured accordingly.

1. I believe the authors’ findings are sufficiently valid although a valid criticism of their experimental design is that the injection of a sham dsRNA control is preferred in knock down experiments rather than the use of a non-RNA (PBS) injection control as used here. This is to eliminate the possibility that the observed reduction in expression of Piwi1 (averaging ~50%) was not simply a toxic off-target effect (admittedly unlikely).
2. In relation to the above, the authors state that the proportion of cells expressing Piwi1 RNA actually rises in the presence of inhibitory dsRNA but they do not clearly discuss this effect or its relevance to the observed phenotype. The proportions are presumably calculated from the total numbers of cells counted in their sections (male and female). Are the authors trying to say that they observe a form of gonadogenic arrest with a gradual accumulation of Piwi1 expressing cells at earlier stages in the process? Presumably, these cells then become apoptotic?
3. The authors should state what statistical test(s) they used to analyse their quantitative data as this could have a bearing on the strength of the significant differences they report.
4. The authors often fail to provide adequate references or details supporting some of their methods. For example, on line 86, they mention ‘common methods’ for paraffin embedding but a reference should be provided nevertheless. Similarly, on line 96, mention is made of Molecular Cloning III, which is presumably a manual but no reference is provided. Again on line 114, the neighbour joining method in MEGA 5 is mentioned but without a reference. The authors also fail to mention who the suppliers are for many of the reagents used including poly-l-lysine, NBT/BCIP, formamide, xylene, ethanol, and many other chemicals. I know it is tedious but suppliers should be provided.
5. The authors should indicate where the consensus PAZ and PIWI domains are on the alignment shown in Figure S1.

Some minor points relating to word usage should be considered as follows:

Line 60 ‘Identifying early GCs is benefit for the study of gametogenesis’ replace 'benefit for' with 'beneficial to'.
Line 80. Change ‘Another parts were fixed’ to ‘Other tissues were fixed.’

Reviewer 3 ·

Basic reporting

No comment.

Experimental design

No comment.

Validity of the findings

No comment.

Additional comments

In this study, the authors identified piwi gene from scallop, and further characterized it through qRT-PCR, knockdown techniques etc. They found that piwi was essential for gametogenesis in the scallop. Overall, the methods are sound, and the findings are interesting and important. But there are several issues need to be addressed before acceptance.
1. The authors need to be more careful for methods description, for instance, In Line 97, "Reverse transcription and qRT-PCR were performed ......and Primescript cDNA amplification kit." I believe the two kits listed was for full length cDNA preparation, but not for qRT-PCR.
2. In Line 189, "Phylogenetic analysis showed that the predicted Cf-piwi first clustered with C. gigas". But from Fig.1, Cf-piwi actually is first clustered with C. gigas and M. galloprovincialis. Also, I suggest to add all species used in phylogenetic analysis in Method section, including the common names and Latin names.
3. For those figures of qRT-PCR results, how can we tell if they are significantly different? The authors says "Different letters indicate statistically significant difference(P<0.05)", but from the figure, we can only see the labels "a, b, c, d", what do those 4 letters represent for?

---

## Round 0.2 · accepted · Accept

· Academic Editor

Accept

I am satisfied with the general response and changes, and the explanation why a scrambled dSRNA control was not used in this instance, which was the main issues raised by one of the reviewers.